# Prolonged Sitting Causes Leg Discomfort in Middle Aged Adults: Evaluation of Shear Wave Velocity, Calf Circumference, and Discomfort Questionaries

**DOI:** 10.3390/jcm11144024

**Published:** 2022-07-12

**Authors:** Kumiko Okino, Mitsuhiro Aoki, Masahiro Yamane, Yoshiaki Kataoka, Asami Nitta, Chikashi Kohmura

**Affiliations:** 1Department of Clinical Laboratory Medicine, School of Clinical Technology, Health Sciences University of Hokkaido, 2-5 Ainosato, Kita-ku, Sapporo 002-8072, Hokkaido, Japan; kokino@hoku-iryo-u.ac.jp (K.O.); ckohmura@hoku-iryo-u.ac.jp (C.K.); 2Department of Physical Therapy, Graduate School of Rehabilitation Science, Health Sciences University of Hokkaido, 1757 Kanazawa, Toubetsu-cho, Ishikari-gun 061-0293, Hokkaido, Japan; 3Department of Physical Therapy, Health Science University Hospital, 2-5 Ainosato, Kita-ku, Sapporo 002-8072, Hokkaido, Japan; m-yamane@hoku-iryo-u.ac.jp (M.Y.); aquarius_plus_150g@yahoo.co.jp (Y.K.); asami-n@hoku-iryo-u.ac.jp (A.N.)

**Keywords:** middle-aged male and female, prolonged sitting, shear wave velocity, leg circumference, leg discomfort

## Abstract

Introduction: Prolonged sitting causes leg discomfort. We evaluated shear wave velocity (SWV) of leg muscles, leg circumference, and leg discomfort associated with 2 h sitting. Methods: Twenty-one middle-aged men and 19 middle-aged women participated in the study. SWV and leg circumference was measured just after sitting, 60 min, 120 min, and after 3 min of leg raising. Leg discomfort was assessed before sitting and 120 min. Results: SWV was significantly greater in men than women and increased over time, and decreased with leg raising. The percentage increase in lower leg circumference was significantly greater in women than in men, and it increased over time. Leg discomfort significantly increased after 120 min in both men and women. Discussions: Because SWV is proportional to an increase in intramuscular compartment pressure in the lower leg, intramuscular compartment pressure increased over time with sitting and decreased with leg raising. Considering the changes in SWV and leg circumference, it was inferred that prolonged sitting causes an increase in intramuscular compartment pressure and intravascular blood volume, as well as an increase in water content in the leg subcutaneous tissue. Leg discomfort was estimated to be due to increased intra-leg fluid. Brief leg raising may resolve leg edema and discomfort.

## 1. Introduction

A significant number of people engage in sedentary behaviors for extended periods of time, and physical inactivity is widespread. Americans spend 55% of their waking hours (7.7 h per day) engaging in sedentary behaviors. Europeans spend 40% of their leisure time (2.7 h per day) watching television [1]. The low level of physical activity is presumably influenced by multiple factors. Among them, environmental factors include traffic congestion, air pollution, lack of parks and trails, and lack of sports and leisure facilities [2]. Television viewing, video viewing, and cell phone use are positively correlated with sedentary lifestyles [3]. Sedentary lifestyles are expected to continue to increase due to this social context [4].

Many of the physical activity-related instructions in clinical settings focus on increasing physical activity levels and not on reducing sedentary behaviors that pose risks to health. In addition to understanding and communicating the impact of sedentary lifestyles on health to patients, health care professionals in various disciplines, including clinicians, need to consider the implications [4]. In this study, the impact of a sedentary lifestyle on health was examined, particularly in the context of occupational leg symptoms due to sedentary work and leg edema in the elderly due to prolonged sitting. Based on the results, we try to propose methods that should be implemented to maintain a healthy lifestyle for those who have a sedentary posture for long periods of time.

### 1.1. Lower Leg Discomfort Symptoms Due to Prolonged Sitting

Winkel 1971 [5] reported that prolonged sitting due to occupational seating positions caused leg discomfort and rated the severity of leg discomfort on an 8-point scale. Widmer 1978 [6] specifically described leg discomfort symptoms as tension, heaviness, swelling, and so on. Recently, lower leg discomfort symptoms associated with long standing and sitting jobs have been reported [7,8], and Saito 2016 [9] performed factor analysis on multiple discomfort symptoms expressed by workers and extracted lower leg discomfort symptoms, which had a large contribution rate. Sudo 2010 [10] reported distinctive lower leg discomfort symptoms associated with working nurses. 

Increased leg water content and fluid retention in subcutaneous tissue are caused due to airline boarding and long-distance bus travel. The development of thrombophlebitis and pulmonary emboli during long-distance flights [11,12] was reported, drawing attention to leg symptoms caused by prolonged sitting. Subsequently, fluid accumulation in the lower legs during long-distance bus rides by measuring limb volume using an optoelectronic scanner system (Perometer^®^) [13] has been reported. The report demonstrated that during 10 h total amount of 105 mL (1.6% of leg volume) was increased during bus rides. Leg edema and venous blood flow in the long-distance flight simulation environment by using an optoelectronic scanner system (Perometer^®^) and sonographic measurement of diameters of femoral, popliteal, and medial gastrocnemius veins [14] have been reported. The report demonstrated that the increase in lower leg volume during 4 h was 109 mL, but there was no significant increase in the diameter of calf veins. Thus, perineal venous hemodynamic plethysmographic analysis of tissue fluid has been used to elucidate the mechanisms of leg edema and leg discomfort symptoms associated with prolonged sitting [14,15,16,17]; few reports have observed changes produced in the lower leg muscles by prolonged sitting.

### 1.2. Prolonged Sitting and Shear Wave Velocity (SWV) Changes Related to Pressure Changes in the Muscle Compartment

Recently, ultrasound-based shear wave elastography has been used to evaluate muscle elastic modulus, a measure of skeletal muscle stiffness, and to quantitatively assess the physical properties of muscle tissue [18,19,20]. We measured the SWV of the lower leg muscles in order to observe changes in stiffness of the muscles during prolonged sitting [21]. Toyoshima (2020) [22] reported that SWV increased in proportion to the increase in pressure in the tibialis anterior compartment using turkey legs. Therefore, the increase in SWV associated with prolonged sitting was found to be due to an increase in the compartment pressure within the lower leg muscle [21].

Leg discomfort symptoms and edema are more common in women than in men, and leg edema increases with age. It has been reported that sitting for 2 h increases leg circumference, with an increase of 1.4% in men versus 2.2% in women [23]. The leg discomfort symptoms and edema that vary by gender are reported to increase with age [24]. Suehiro 2014 and Sato 2015 [7,25] reported that leg edema with increased leg circumference was found in about half of elderly patients. Iuchi 2017 [26] reported the usefulness of both ultrasound echo intensity measurement of subcutaneous connective tissue and leg circumference measurement in assessing leg edema in elderly patients.

Thus, it has been shown that leg edema and discomfort symptoms appear with prolonged sitting, but there are no reports observing leg muscle stiffness changes, leg edema, and discomfort symptoms in elderly patients. The purpose of this report was to evaluate the SWV of the lower leg muscles, leg circumference, and leg discomfort symptoms following 2 hours of prolonged sitting in order to determine the relationship between prolonged sitting and the development of lower leg discomfort symptoms.

The hypothesis of this report is that 2 h of prolonged sitting significantly increases SWV over time in middle-aged men and women and decreases with lower leg raising. SWV in men is greater than in women. The rate of increase in leg circumference with sitting significantly increased over time and decreased with raising of the lower leg. The value is significantly greater in women than in men. Leg discomfort symptoms significantly increased after 120 min sitting in both men and women. 

Based on these results, the cause of leg discomfort symptoms due to prolonged sitting is estimated from the results of SWV and leg circumference measurements.

## 2. Methods 

### 2.1. Subjects

Twenty-one healthy middle-aged men (mean age 56.2 ± 12.0 years) and 19 middle-aged women (mean age 52.9 ± 9.7 years) participated in the study. No difference in age was observed between the two groups. Subjects were recruited via posters posted on bulletin boards of the hospital and university. From the interview, hypertensive subjects with a maximum or minimum blood pressure of 140 mmHg or 90 mmHg or higher, patients with cardio-cerebrovascular disorders, patients with varicose veins of the lower extremities, and those with blood disorders or traumatic diseases of the lower extremities were excluded.

Body mass and body composition of middle-aged men and middle-aged women measured with a Tanita body composition analyzer (MC-180, Tanita Corporation, Tokyo, Japan) are shown in Table 1. Measurements were taken between 3:00 pm and 6:00 pm. Height, weight, BMI, water content, lower limb muscle mass, and lower limb muscle mass per body weight were significantly greater in men than in women (*p* < 0.01, *p* < 0.05 for BMI), while body fat and body water content showed no differences between the two groups. Body fat percentage was significantly greater in women than in men (*p* < 0.05).

The purpose of the study, management of personal information, prohibition of use for other purposes, and free participation in the study were explained. Consent to participate in the study was also obtained in writing. Informed consent was obtained for the release of data excluding personal information, including images obtained. In accordance with the Declaration of Helsinki, the subjects’ personal rights and data protection were explained. This study was conducted with the approval of the University Research Ethics Review Committee (approval number 19R115109).

### 2.2. Sample Size

To determine the sample size for SWV and leg circumference analysis by split-plot analysis of variance, G*power (Version 3.1.9.2 Heinrich Heine University, Düsseldorf, Germany) was used, where Cohen’s moderate effect size of 0.25, 2 groups, 4 levels, an alpha value of 0.05, a beta value of 0.8, and since the interlevel intraclass correlation coefficient (ICC) for measurements is less than 0.2 in our obtained data, 0.2 was used for ICC [27]. Then sample size was calculated. As a result, it was 14 cases. For the SWV measurement in our study, the sample size was 21 middle-aged males and 19 middle-aged females, and for the leg circumference measurement, the sample size was 15 middle-aged males and 16 middle-aged females due to missing values.

To determine the sample size for leg discomfort symptoms by the Wilcoxon signed-rank test with two corresponding groups, the mean and standard deviation for middle-aged men and middle-aged women and the number of interlevel intraclass correlation coefficients were input to calculate the effect size. Using the calculated effect size, the sample size was calculated by G*power with an alpha value of 0.05 and a beta value of 0.8 as inputs. As a result, the sample size for men was calculated to be 9 for women and 11 for men. The sample size for this study was 21 middle-aged men and 19 middle-aged women.

### 2.3. Seated Posture

Subjects sat in a reclining chair and maintained a sitting posture for 2 h in a relaxed position, with joint flexion angles measured with a goniometer to maintain 70 degrees of hip flexion, 80 degrees of knee flexion, and 20 degrees of ankle flexion. Ankle joint flexion was set at 20 degrees of flexion beyond the relaxation angle at which the triceps muscle relaxes [28,29]. In this limb position, the tibialis anterior muscle was slightly stretched, and the triceps femoris muscle was relaxed. Subjects sat with their left and right legs shoulder-width apart and placed their feet on a weight-bearing stand so that the load on both soles was equal. They remained relaxed during the measurement, and SWV, lower leg circumference, and lower leg discomfort symptoms were measured (Figure 1a,b).

### 2.4. Measurement of SWV

Using an ultrasound system (Aplio 500 Canon, Tokyo, Japan) and a 6 cm wide linear probe (PLT100BT5, Canon, Tokyo, Japan), SWV of the lateral gastrocnemius, medial gastrocnemius, soleus, and tibialis anterior muscles of the left lower limb was measured immediately after the start of sitting, 60 min later, and 120 min later. The left lower limb was then elevated for 3 minutes with the knee joint extended in the recliner sitting position and then returned to the sitting position for measurement (Figure 2). For individual subjects, probes were applied to the medial and lateral gastrocnemius, soleus, and tibialis anterior muscle bellies at designated positions (medial and lateral gastrocnemius, 30% proximal lower extremity sites; middle of soleus, central lower extremity site; tibialis anterior, 30% proximal lower extremity site) [19,21]. A rectangular skin mark was affixed with adhesive tape to the most prominent part of the muscle belly before measurement, and the probe was placed on the rectangular mark (Figure 3a,b). The direction of the gastrocnemius and soleus muscle fibers run obliquely due to feathering structures, so the long axis of the probe was aligned with the anatomical direction of shortening of the muscle fibers. Measurements were performed by an experienced clinical laboratory technician (K.O.). The temperature and humidity in the laboratory were maintained at 25 °C and 30–40%, respectively. Measurements of SWV were taken between 3:00 pm and 6:00 pm.

### 2.5. Accuracy of SWV

We refer to a report in which measurements were made with an ultrasound system (Aplio 500, Canon, Tokyo, Japan) and a linear probe (PLT100BT5, Canon, Tokyo, Japan) with a field of view of 58 mm and a standard operating frequency of 10 MHz (maximum 14 MHz), and measurement accuracy was evaluated using phantoms [30]. According to the report, the accuracy of the velocity measurement by Aplio500 using a phantom (CIRS, Norfork, Virginia, USA; model 049 and 049A) of 8.0 ± 3.0 kPa is 0.49 kPa, and the precision (Coefficient of Variation and Confidence Interval) is 6.96%, 5.79–8.13%. It is known that the accuracy and precision of SWV measurements in the range of 2–6 cm depth of phantom material do not differ with depth [30]. In their study, validation of SWV measured by 5 major ultrasound systems (VTQ, VTIQ, EPIQ 5, Aixplorer, and Aplio 500) was performed using various phantoms with mutual comparison. The amount of measurement errors by Aplio 500 was minimum among them. The range of leg muscle SWV measured by the Aplio 500 in this experiment was 1.0–3.0 m/s (3.0–9.0 kPa), so the reliability and validity of the measurement of the SWV of the leg muscles by Aplio 500 was assured [21].

### 2.6. SWV Measurements

Ultrasound images were recorded along the long axis of each muscle. The probe was scanned parallel to the superficial fascia of each muscle. The region of interest (ROI) size was set at 10 × 10 mm^2^. The analysis region (target ROI) was set as a 5 mm diameter circle at the center of the ROI [31] (Figure 4). Ultrasound images were taken three times each. The mean value was calculated from the three images obtained from each muscle. The SWV of the gastrocnemius and soleus muscles was measured with the ankle joint flexed 20 degrees, and the muscles relaxed [32]. SWV of the tibialis anterior muscle was measured with the ankle joint flexed 20 degrees and 10 degrees beyond the relaxation angle. Considering both longitudinal measurements and muscle length, both muscles were considered isotropic for this measurement, and the elastic modulus was calculated using the following equation:E = 3ρVs^2^

### 2.7. Reliability Analysis of SWV Measurements

The SWV measurements conducted in this report were performed by the same inspector, measuring equipment, measuring conditions, and measuring site as in the previous report. Therefore, the reliability of the present measurement was determined to be consistent with that of the previous report. According to the results, the intra-day intra-inspector reliability (1, 1) and coefficient of variation (CV) were 0.88% and 3.9% for the lateral gastrocnemius muscle, 0.87% and 5.7% for the medial gastrocnemius muscle, 0.92 and 7.5% for the soleus muscle, and 0.92% and 7.5% for the tibialis muscle. The inter-day intra-examiner reliability (1, 3) and coefficient of variation (CV) were 0.77% and 3.7% for the lateral gastrocnemius, 0.82% and 6.5% for the medial gastrocnemius, 0.76% and 8.7% for the soleus muscle, and 0.81% and 12.0% for the tibialis anterior muscle [21].

### 2.8. Surface Electromyography Measurement

Active wireless surface electrodes (Nihon Kohden Corporation, Tokyo, Japan) were placed to check the contraction of the lower limb muscles of the subject sitting on a reclining seat and the muscle activity of the medial and lateral gastrocnemius, soleus, and tibialis anterior muscles were observed. The placement sites of the surface electrodes were adjusted according to Hermens et al. [33]. The sites of electromyography electrode placement were cleaned with alcohol cotton, and the stratum corneum was removed with paste. EMG was recorded at 2000 Hz, collected on a Web-1000 (Nihon Kohden, Tokyo) polygraph system, and analyzed using LabChart version 7 (AD Instruments, Tokyo). EMG data during shear wave transmission velocity measurements were continuously displayed to the examiner and subject on a PC screen to promote relaxation of the subject. The appearance of muscle activity was determined by observing action potentials above the baseline 2SD amplitude. Preliminary experiments determined that no muscle contractions were observed in the lower leg muscles due to reclining seat seating under the conditions of this measurement [21].

### 2.9. Lower Leg Circumference Measurement

The maximum circumference of the left lower leg was measured over time according to the standard lower leg circumference measurement method reported in JARD 2001 [34,35] (Figure 5). The reliability analysis was performed for limb circumference by Labs 2000 [36]. The measurement has been reported to diagnose 87.8% of cases of edema of the lower leg, especially in the elderly [26]. 

### 2.10. Lower Leg Discomfort Symptom Assessment

Occupational lower leg symptoms were initially rated on an 8-point scale of lower leg discomfort (Winkel 1981) [5]. Furthermore, Widmer 1978 [6] reported that among 4529 sedentary workers, 44% of men and 70% of women complained of leg discomfort symptoms such as tension, heaviness, and swelling. Saito 2016 [9] extracted items with a large contribution rate out of 17 lower leg discomfort symptoms by factor analysis. In this report, we created a 6-item rating of leg discomfort symptoms (tension, heaviness, swelling, numbness, pain, and heat) by taking into account the survey items of Widmer 1978 [6] and Saito 2016 [9] (Figure 6). Each item was rated on an 11-point scale from 0 to 10. Leg discomfort symptom ratings were made before the start of sitting and after 120 min.

### 2.11. Statistical Analysis

Due to the normality of the Shapiro-Wilks test for body size and composition of the subjects, an uncorrelated t-test was used. Due to the normality of the Shapiro-Wilks test for SWV in the four muscle groups and leg circumference, SWV measurements and leg circumference values obtained over time in two groups, middle-aged men and women, were analyzed by a split-plot analysis of variance. The Shapiro-Wilks test showed no normality for leg discomfort symptoms, so a Wilcoxon signed-rank test with two corresponding groups was used. Statistical analysis was performed with R4.02, and significance levels of 0.05 were used. 

## 3. Results 

### 3.1. SWV 

With the exception of the lateral gastrocnemius, the SWV of the medial gastrocnemius, soleus, and tibialis anterior muscles of middle-aged men was significantly greater than that of middle-aged women (*p* < 0.01). In all lower leg muscles of middle-aged men and middle-aged women, SWV was significantly greater at 60 and 120 mins than immediately after the start of the sitting and significantly decreased after lower leg raising (*p* < 0.001) (Figure 7 MG LG SOL TA, Table 2). 

The rate of change in lower leg circumference was significantly greater in middle-aged women than in middle-aged men (*p* < 0.05) (Figure 8, Table 3). The rate of change in lower leg circumference significantly increased after 60 and 120 min of sitting and decreased with raising of the lower leg for both men and women (*p* < 0.001).

### 3.2. Lower Leg Discomfort Symptom Assessment

No differences in lower leg discomfort symptoms were found between middle-aged men and middle-aged women. They significantly increased after sitting compared to before sitting in both men and women (*p* < 0.001) (Figure 9a). Of the six lower leg discomfort symptoms, tension, swelling, heaviness, and numbness markedly increased (Figure 9b).

## 4. Discussions

### 4.1. Changes in SWV

The results of our previous study, in which SWV of the leg muscles increased over time in young adult males in a 2 h sitting [21], were comparable over time to our results, which were measured in middle-aged men and women. Because SWV is proportional to an increase in intramuscular compartment pressure in the lower leg [22], our results of increased SWV over time in both middle-aged men and women were presumed to be due to increased pressure within the lower leg muscle compartment. The rapid decrease in SWV with leg raising is presumably due to a decrease in intra-compartmental pressure as intramuscular venous blood is drained from the leg muscle and the intramuscular compartment volume is reduced. 

The higher SWV in middle-aged men than in middle-aged women suggests that the pressure in the leg muscle compartment over time is greater in men. This suggests that in the sitting position with muscle relaxation, the pressure in the lower leg muscle compartment was initially higher in men. Although blood pressure was not measured in this study, men’s blood pressure was higher than women’s blood pressure, suggesting that intramuscular compartment pressure in men may have been higher from the beginning. According to a survey by Miura 2013 [37], Japanese systolic blood pressure is 137.2 mmHg for men and 129.7 mmHg for women in their 50s.

### 4.2. Leg Circumference

It is known that prolonged sitting increases lower leg volume and circumference, which is assumed to be caused by the accumulation of subcutaneous fluid [8,13,14,23]. The results of this study showed a significant increase in leg circumference in middle-aged men and middle-aged women, suggesting that the 120-min prolonged sitting caused the accumulation of blood in the leg veins and subcutaneous tissue fluid in both men and women [9,21]. The greater rate of leg circumference increase in middle-aged women compared to middle-aged men was assumed to be due to the fact that women tend to store more fluid in a large subcutaneous tissue than men.

### 4.3. Reduction of SWV and Leg Circumference by Lower Leg Raising

Leg raising and leg muscle active contraction have been recommended for the reduction of leg edema that occurs during prolonged standing or sitting [15,38]. The muscle pumping effect has been reported to produce the compression of the sinus vein in the soleus muscle by active muscle contraction and the drainage of stagnant venous blood from the veins [39]. On the other hand, the reduction in hydrostatic pressure due to leg raising promotes the outflow of blood in the leg veins and the redistribution of venous blood. However, no specific measurement index has been provided to determine how much venous blood is drained or how much the muscle compartment pressure changes after the lower leg raising. Our measurements revealed that 3 min of leg raising significantly decreased all leg muscle SWV and decreased intramuscular compartment pressures, suggesting a decrease in intramuscular leg compartment volume associated with rapid venous blood drainage by leg raising. This phenomenon is consistent with the significant reduction in leg circumference resulting from leg raising. Since the SWV is expected to increase slowly and the leg circumference will also increase gradually, brief leg raising may be a simple and useful means of reducing leg edema. Elevating the lower extremity at least every hour may avoid an increase in intramuscular compartment pressure. 

### 4.4. Body Composition and Lower Limb Circumference

The body water and body fat content of the subjects were consistent with age-specific measurements of the Japanese population as measured by the impedance method [40]. According to the body composition measurements made before the start of the sitting in this report, body water content standardized by body weight showed no difference by gender, suggesting that there is no difference in leg water content between women and men. In addition, body fat percentage is significantly greater in women than in men, and it is estimated that the percentage of leg adipose tissue volume in women is greater than in men. It is presumed that during prolonged sitting, a greater rate of increase in leg circumference measured in women is due to large extracellular fluid retention by a greater percentage of leg adipose tissue mass. 

### 4.5. Assessment of Lower Leg Discomfort Symptoms

The reason for lower leg discomfort symptoms during prolonged sitting or standing has been reported to be due to swelling of the lower leg, as indicated by an increase in lower leg volume and lower leg circumference [9,10,12,13]. Our previous study of SWV in young adult males estimated that prolonged sitting causes increased pressure within the lower leg muscle compartment and water retention in the lower leg subcutaneous tissue [21]. The present results demonstrated that 120 min of sitting significantly increased lower leg discomfort symptoms in both middle-aged men and women, indicating that the increase in leg discomfort symptoms is due to an increase in water volume in the lower leg. The significant increase in tension, swelling, heaviness, and numbness out of the six lower leg discomfort symptoms indicated that these four discomfort symptoms were subjective symptoms suggestive of the development of lower leg edema.

### 4.6. Sedentary Lifestyle Linked to Musculoskeletal Disease

A report analyzing the correlation between chronic knee pain and daily sitting time (5–7 h, 8–10 h, and more than 10 h) found that the incidence of chronic knee pain was higher the more sitting time per day. In particular, sitting time of more than 10 h per day was significantly correlated with chronic knee pain (adjusted OR, 1.28; 95% CI, 1.02-1.61; *p* = 0.03). The study recommended less than 10 h per day of sitting time [41]. Osteoarthritis of the knee is a leading musculoskeletal disease that reduces the mobility of the elderly [42] and keeps patients engaged in sedentary behavior. Knee cartilage is nourished by joint fluid, and it is known that cartilage damage occurs when cartilage compression is sustained due to joint immobility [43], supporting the result that prolonged sitting produces knee joint pain. It is also known that leg edema frequently occurs in cases of knee osteoarthritis, and it is presumed that joint immobility due to prolonged sitting in the elderly leads to the development of leg edema as well as knee joint pain. In these cases, brief lower-leg elevation every 1–2 h may simultaneously prevent cartilage damage and leg edema by releasing mechanical compression of the articular cartilage and restoring joint fluid circulation through knee joint movement, and by reducing intramuscular compartment pressure through drainage of blood in the leg veins.

### 4.7. Future Study

Prolonged sedentary activity time worsens outcomes of lifestyle-related diseases, malignancies, cardiovascular disorders, occupational leg edema, thrombophlebitis, and knee osteoarthritis with leg edema in the elderly. However, even within the same sitting period, patterns of sitting time vary in daily life [44]. Non-invasive ultrasound-based screening studies of lower leg muscle groups are required to measure prolonged sitting time patterns that can avoid leg health problems, e.g., continuous sitting without rest, intermittent sitting with standing and walking, and sitting interspersed with lower extremity exercise.

### 4.8. Research Limitations

#### 4.8.1. Measurement of Muscle at Rest

In our study, we measured the leg immobile in the sitting position for a long time. However, except in special cases, it is rare to have a situation in which the subjects do not move their legs for 2 h during sitting. In the future, it is necessary to make ultrasound measurements that take the motion of the lower limbs into account.

#### 4.8.2. Measurement up to 2 h

In this study, we adopted the lower leg sitting condition for up to 2 h. According to previous reports, the occurrence of economy class syndrome is known to occur after a flight longer than 6 h. While avoiding the development of venous thrombosis, measurements should also be performed under conditions exceeding 2 h [19,39]

No direct intramuscular pressure measurement is performed.

In this measurement, the internal pressure of the leg muscle compartment was estimated as the change in the value of the SWV. For more direct measurement, intramuscular pressure measurement is required [45,46].

#### 4.8.3. Venous Hemodynamics

In the present measurement, blood retention in the vein was not directly assessed. Since ultrasonic measuring devices can measure the increase in the diameter of the intramuscular veins and blood flow, there is a need to investigate the hemodynamics of venous blood in conjunction with the measurement of SWV [14,22,47].

## 5. Conclusions

Considering the changes in SWV and leg circumference associated with prolonged sitting, it is estimated that sitting legs cause an increase in intramuscular compartment pressure and an increase in intravascular blood volume with retention of subcutaneous fluid in the lower leg. Raising the lower leg decreases pressure and volume in the muscle compartment and water retention. Leg discomfort symptoms were significantly increased by prolonged sitting, which was presumed to be caused by increased fluid in the lower leg. Brief leg raising was found to be an effective means of intra-compartmental pressure reduction by draining leg venous blood, which may resolve leg edema and discomfort.

## Figures and Tables

**Figure 1 jcm-11-04024-f001:**
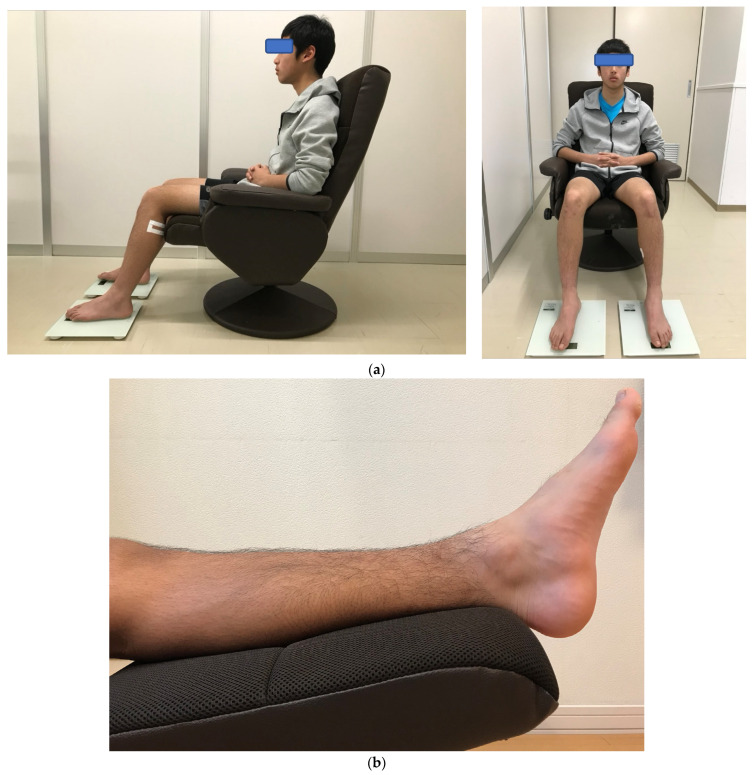
Sitting posture and leg raise appearance. (**a**) Sitting position on a reclining chair. (**b**) Leg raise on a stool. Permission with Plos One 2021 [21].

**Figure 2 jcm-11-04024-f002:**
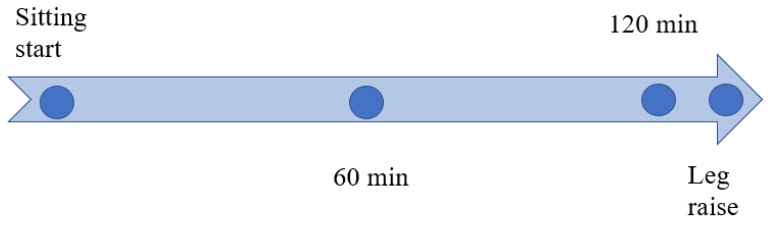
Experimental design, time point for data acquisition. Permission with Plos One 2021 [21].

**Figure 3 jcm-11-04024-f003:**
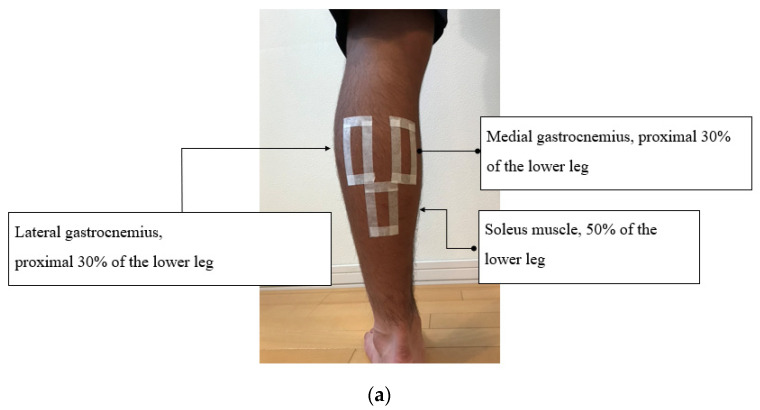
Left lower leg of a participant. (**a**) Sites of recording of shear wave velocity at lateral, medial gastrocnemius, and soleus muscles. (**b**) Sites of recording of shear wave velocity at tibialis anterior muscle. Permission with Plos One 2021 [21].

**Figure 4 jcm-11-04024-f004:**
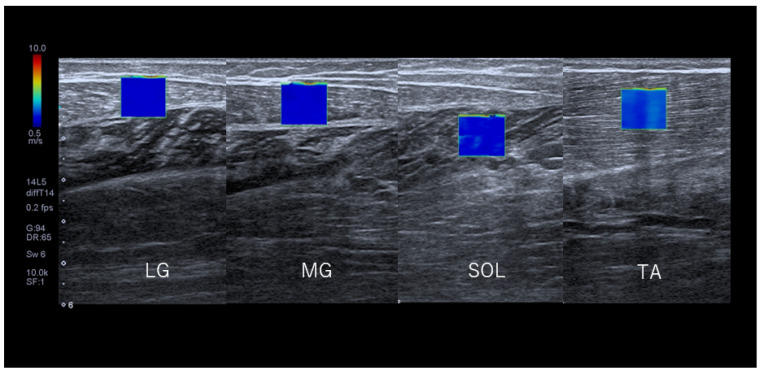
Image of shear wave elastography of lower leg muscles. LG: Lateral gastrocnemius, MG: Medial gastrocnemius; SOL: Soleus; TA: Tibialis anterior. ROI: Region of interest: 1 cm × 1 cm. Permission with Plos One 2021 [21].

**Figure 5 jcm-11-04024-f005:**
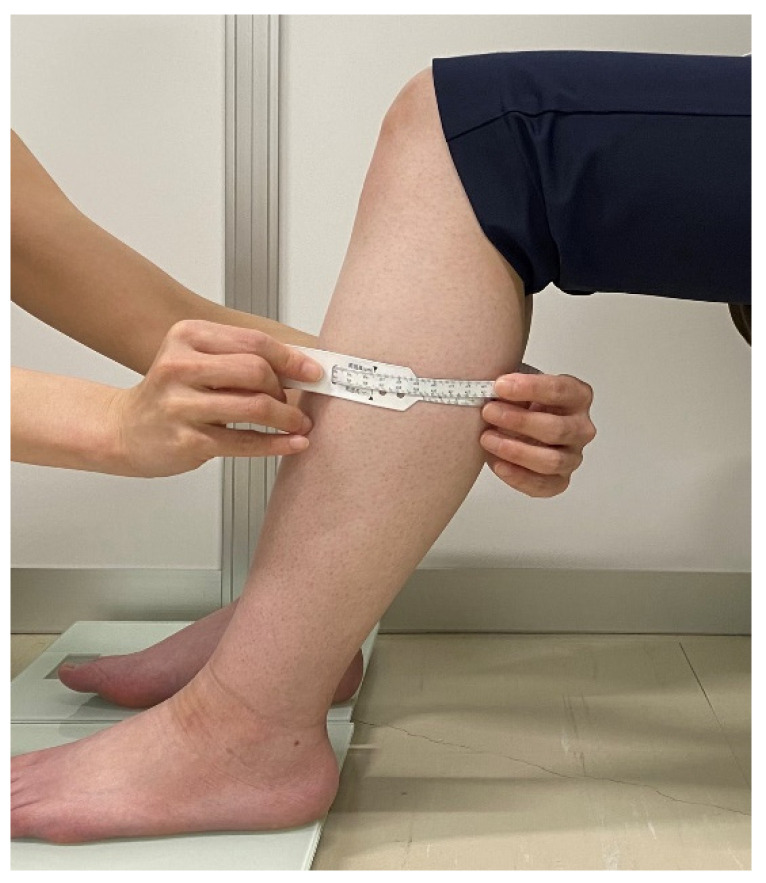
Measurement of calf circumference. The thickest part of the lower leg was measured three times, and the average value was calculated.

**Figure 6 jcm-11-04024-f006:**
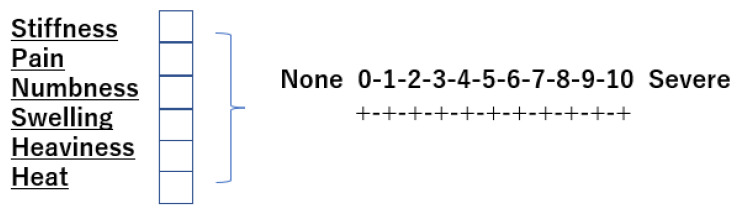
Evaluation sheet of lower leg discomfort. The eleven point rating scale used in the field study to measure the perception of discomfort in the leg. Six items of discomfort on the inspection sheet.

**Figure 7 jcm-11-04024-f007:**
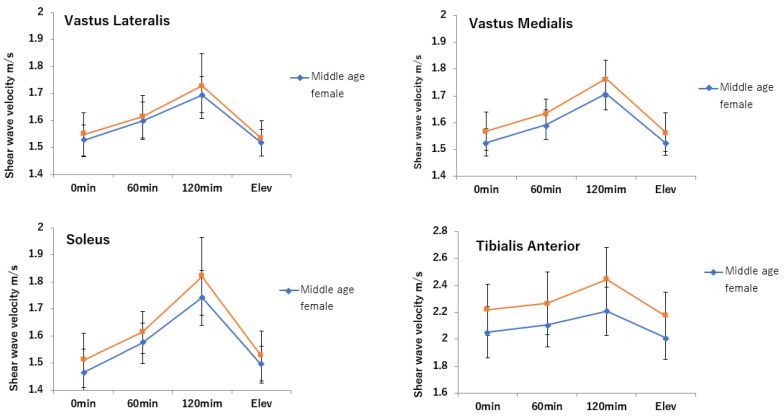
Shear wave velocity of LG, MG, SOL, and TA over time. Unit: m/s. Average ± SD. LG: Lateral gastrocnemius; MG: Medial gastrocnemius; SOL: Soleus; TA: Tibialis anterior.

**Figure 8 jcm-11-04024-f008:**
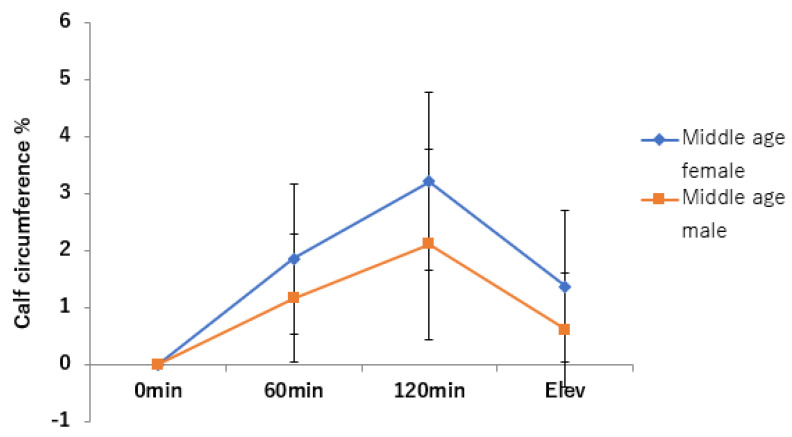
Change in calf circumference over time.

**Figure 9 jcm-11-04024-f009:**
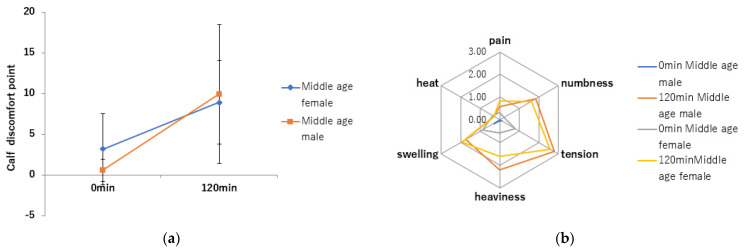
Lower leg symptom discomfort assessment. (**a**): calf discomfort points. (**b**): radar chart of six items of calf discomfort.

**Table 1 jcm-11-04024-t001:** Demographic data of middle aged male and female participants.

	Height	Weight	BMI	Body Fat	Body Water	Leg Muscle	Body Fat	Body Water	Leg Muscle
	(m)	(kg)		(kg)	(kg)	(kg)	(%)	%	(%)
Middle age male	168 (6.5)	65.6 (8.5)	23.6 (5.3)	14.3 (5.3)	35.8 (4.4)	8.7 (1.1)	21.8 (7.4)	54.7 (5.2)	13.1 (0.9)
(n = 21)									
Middle age female	156.6 (6.5)	52.1 (7.5)	21.4 (3.2)	15.2 (6.5)	27.0 (2.5)	6.0 (0.5)	27.1 (7.6)	52.3 (5.2)	11.7 (1.3)
(n = 19)									
Mean (SD)									

**Table 2 jcm-11-04024-t002:** Shear wave velocity of LG, MG, SOL, and TA over time. Unit: m/s. Average ± SD. LG: Lateral gastrocnemius; MG: Medial gastrocnemius; SOL: Soleus; TA: Tibialis anterior.

		LG	MG	SOL	TA
Middle age male	0 min	1.55 (0.08)	1.57 (0.07)	1.51 (0.10)	2.22 (0.19)
(n = 21)	60 min	1.61 (0.08)	1.63 (0.05)	1.61 (0.08)	2.27 (0.23)
	120 min	1.73 (0.12)	1.76 (0.07)	1.82 (0.14)	2.44 (0.24)
	Leg raising	1.53 (0.06)	1.56 (0.07)	1.53 (0.09)	2.17 (0.18)
Middle age female	0 min	1.53 (0.06)	1.53 (0.05)	1.46 (0.09)	2.05 (0.19)
(n = 19)	60 min	1.60 (0.07)	1.59 (0.06)	1.57 (0.08)	2.11 (0.16)
	120 min	1.70 (0.07)	1.71 (0.06)	1.74 (0.10)	2.21 (0.18)
	Leg raising	1.52 (0.05)	1.52 (0.05)	1.49 (0.07)	2.01 (0.16)
Unit: m/s					
Mean (SD)					

**Table 3 jcm-11-04024-t003:** Change of leg circumference over time. Relative calf circumference: Circumference values of 60 min, 120 min, and Leg raising are divided by a circumference value of 0 min.

		Increase in Calf Circumference
Middle age male	0 min	0 (0)
(n = 15)	60 min	1.16 (1.13)
	120 min	1.85 (1.52)
	Leg raising	0.61 (1.00)
Middle age female	0 min	0 (0)
(n = 16)	60 mis	1.86 (1.32)
	120 min	3.20 (1.55)
	Leg raising	1.37 (1.32)
Relative calf circumference: Circumference values of 60 min, 120 min, and Leg raising/Circumference value of 0 min		
Unit: %		
Mean (SD)		

## Data Availability

The data presented in this study are openly available in Text, Figures, and Tables.

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
