# Peer review of "Prolonged Sitting Causes Leg Discomfort in Middle Aged Adults: Evaluation of Shear Wave Velocity, Calf Circumference, and Discomfort Questionaries"

_jcm, 2022, doi:10.3390/jcm11144024_

Round 1
Reviewer 1 Report
The authors have conducted an interesting study with clinical potential. Some concerns need to be addressed.
1. In the section "Increased leg water content and fluid retention in connective tissue due to airline boarding and 40 long-distance bus travel" the authors should rather synthesize the findings of the reported studies rather than presenting their methods
2. The discussion should include a paragraph presenting the limitations of the study and one paragraph elaborating on future research. With regard to the latter it is important to discuss the key lessons that healthcare professionals can draw from this study. Are there any implications to clinical practice or to everyday life activities? Which are the knowledge gaps that should be addressed by future studies?
3. Sedentary behavior is a part of the contemporary lifestyle (and in many cases of the contemporary working routine) that affects other parts of the musculoskeletal system and other organ systems than the lower extremities. The authors should bring this perspective into the discourse as well, perhaps future research should delve into different patterns of sitting to determine which one is less burdenful for the lower extremities - more information on the matter can be found in https://www.sciencedirect.com/science/article/pii/S2666548422000117 and https://www.kjfm.or.kr/journal/view.php?doi=10.4082/kjfm.20.0165 (many more relevant studies are published, up to the discretion of the authors to search relevant resources)
4. The manuscript's language should be proofread and improved. The most striking example is "siting" (sic) in the title. To the best of my knowledge, it is written with a double "t" - the rest of the manuscript follows this rule as well.
Author Response
Comments and Suggestions for Authors
The authors have conducted an interesting study with clinical potential. Some concerns need to be addressed.
- In the section "Increased leg water content and fluid retention in connective tissue due to airline boarding and 40 long-distance bus travel" the authors should rather synthesize the findings of the reported studies rather than presenting their methods
Reply: The suggested part was revised by presenting methods and results of measuring leg conditions.
The development of thrombophlebitis and pulmonary emboli during long-distance flights [11, 12] was reported, drawing attention to leg symptoms caused by prolonged sitting. Subsequently, fluid accumulation in the lower legs during long-distance bus rides by measuring limb volume using an optoelectronic scanner system (Perometer®) [13] has been reported. The report demonstrated that during 10 hours total amount of 105 ml (1.6% of leg volume) was increased during bus rides. Leg edema and venous blood flow in the long-distance flight simulation environment by using optoelectronic scanner system (Perometer®) and sonographic measurement of diameters of femoral, popliteal and medial gastrocnemius veins [14] have been reported. The report demonstrated that increase in lower leg volume during 4 hours was 109 ml, but there were no significant increase in diameter of calf veins. Thus, perineal venous hemodynamic plethysmographic analysis of tissue fluid has been used to elucidate the mechanisms of leg edema and leg discomfort symptoms associated with prolonged sitting [14-17], few reports have observed changes produced in the lower leg muscles by prolonged sitting.
- The discussion should include a paragraph presenting the limitations of the study and one paragraph elaborating on future research. With regard to the latter it is important to discuss the key lessons that healthcare professionals can draw from this study. Are there any implications to clinical practice or to everyday life activities? Which are the knowledge gaps that should be addressed by future studies?
Reply: Research Limitation was added in the last part of Discussion
Research Limitations
Measurement of muscle at rest
In our study, we measured the leg immobile in the sitting position for a long time. However, except in special cases, it is rare to have a situation in which the subjects do not move his or her legs for 2 hours during sitting. In the future, it is necessary to make ultrasound measurements that take the motion of the lower limbs into account.
Measurement up to 2 hours
In this study, we adopted the lower leg sitting condition for up to 2 hours. According to previous reports, the occurrence of economy class syndrome is known to occur after a flight longer than 6 hours. While avoiding the development of venous thrombosis, measurements should also be performed under conditions exceeding 2 hours [19, 40].
No direct intramuscular pressure measurement is performed.
In this measurement, the internal pressure of the leg muscle compartment was estimated as the change in the value of the shear wave velocity. For more direct measurement, intramuscular pressure measurement is required [46,47].
Venous Heamodynamics
In the present measurement, blood retention in the vein was not directly assessed. Since ultrasonic measuring devices can measure the increase in the diameter of the intramuscular veins and blood flow, there is a need to investigate the hemodynamics of venous blood in conjunction with the measurement of SWV [14, 22,48].
Reply: Implication to clinical practice and future study was added in the last part of Discussion.
Sedentary Lifestyle Linked to Musculoskeletal Disease
A report analyzing the correlation between chronic knee pain and daily sitting time (5-7 hours, 8-10 hours, and more than 10 hours) found that the incidence of chronic knee pain was higher the more sitting time per day. In particular, sitting time of more than 10 hours per day was significantly correlated with chronic knee pain (adjusted OR, 1.28; 95% CI, 1.02-1.61; P=0.03). The study recommends less than 10 hours per day of sitting time [42]. Osteoarthritis of the knee is a leading musculoskeletal disease that reduces mobility of the elderly [43], and keep patients in sedentary behavior. Knee cartilage is nourished by joint fluid, and it is known that cartilage damage occurs when cartilage compression is sustained due to joint immobility [44], supporting the result that prolonged sitting produces knee joint pain. It is also known that leg edema frequently occurs in cases of knee osteoarthritis, and it is presumed that joint immobility due to prolonged sitting in the elderly leads to the development of leg edema as well as knee joint pain. In these cases, brief lower-leg elevation every 1-2 hours may simultaneously prevent cartilage damage and leg edema by releasing mechanical compression of the articular cartilage and restoring joint fluid circulation through knee joint movement, and by reducing intramuscular compartment pressure through drainage of blood in the leg veins.
Future Study
Prolonged sedentary activity time worsens outcomes of lifestyle-related diseases, malignancies, cardiovascular disorders, occupational leg edema, thrombophlebitis, and knee osteoarthritis with leg edema in the elderly. However, even within the same sitting period, patterns of sitting time vary in daily life [45]. Non-invasive ultrasound-based screening studies of lower leg muscle groups are required to measure prolonged sitting time patterns that can avoid leg health problems, e.g., continuous sitting without rest, intermittent sitting with standing and walking, and sitting interspersed with lower extremity exercise.
- Sedentary behavior is a part of the contemporary lifestyle (and in many cases of the contemporary working routine) that affects other parts of the musculoskeletal system and other organ systems than the lower extremities. The authors should bring this perspective into the discourse as well, perhaps future research should delve into different patterns of sitting to determine which one is less burdenful for the lower extremities - more information on the matter can be found in https://www.sciencedirect.com/science/article/pii/S2666548422000117 and https://www.kjfm.or.kr/journal/view.php?doi=10.4082/kjfm.20.0165 (many more relevant studies are published, up to the discretion of the authors to search relevant resources)
Reply: In the Introduction, sedentary behavior in the perspective contemporary lifestyle is stated based on presented articles. In the Discussion, clinical implication about the musculoskeletal system and future study is stated.
A significant number of people engage in sedentary behaviors for extended periods of time, and physical inactivity is widespread. Americans spend 55% of their waking hours (7.7 hours per day) in sedentary behaviors. Europeans spend 40% of their leisure time (2.7 hours per day) watching television [1]. The low level of physical activity is presumably influenced by multiple factors. Among them, environmental factors include traffic congestion, air pollution, lack of parks and trails, and lack of sports and leisure facilities [2]. Television viewing, video viewing, and cell phone use are positively correlated with sedentary lifestyles [3]. Sedentary lifestyles are expected to continue to increase due to this social context [4].
Many of the physical activity-related instructions in clinical settings focus on increasing physical activity levels and not on reducing sedentary behaviors that pose risks to health. In addition to understanding and communicating the impact of sedentary lifestyles on health to patients, health care professionals in various disciplines, including clinicians, need to consider the implications [4]. In this study, the impact of sedentary life style on health was examined, particularly in the context of occupational leg symptom due to sedentary work and leg edema in the elderly due to prolonged sitting. Based on the results, we try to propose methods that should be implemented to maintain a healthy lifestyle for those who have a sedentary posture for long periods of time.
- The manuscript's language should be proofread and improved. The most striking example is "siting" (sic) in the title. To the best of my knowledge, it is written with a double "t" - the rest of the manuscript follows this rule as well.
Reply: Tipo errors and word inconsistency is addressed correctly, at Title.
Submission Date
05 May 2022
Date of this review
30 May 2022 09:53:51
Reviewer 2 Report
Dear authors,
Thank you for this paper.
I have doubts about the study's methodology, and certain important issues need to be resolved.
1. Title: siting à sitting
2. In the purpose of this study, it would be better to measure intramuscular pressure rather than shear wave velocity.
3. "velocity" and "speed" are mixed throughout the paper. The title and table of the paper states, "shear wave speed" and the text states "shear wave velocity". It must be unified into one.
4. Sample size part (the number of participants) should be in the “subjects” section in methodology.
5. It would be better to present the reliability and validity of the "Ultrasound system (Aplio 500 Canon) and 6ì¸ wide linear probe (PLT100BT5)" equipment and measurement method.
6. Line 97. "Table 1" is marked as duplicate.
7. Why did you distinguish "Shear Wave Velocity" from "The Rate of Change in Shear Wave Velocity" in the results? The two do not seem to mean much differently. Also, even the table presented in the conclusion is the same.
8. Line 124: “a, 1b” à “1a, 1b” & Line 125, 126: a à 1a, b à 1b
9. Line 150: “a. 3b” à “3a, 3b” & Line 151, 152: a à 3a, b à 3b
10. In the conclusion section, raising of the lower leg decreases the intra-partial pressure by reducing the "connect issue water". However, this was not sufficiently presented in the discussion. It was presented only on line 366-368. This is too reasoning and leap forward to draw the conclusions presented by the author.
Author Response
Comments and Suggestions for Authors
Dear authors,
Thank you for this paper.
I have doubts about the study's methodology, and certain important issues need to be resolved.
- Title: siting à sitting
Reply: Tipo error and word inconsistency is addressed correctly, at Title.
- In the purpose of this study, it would be better to measure intramuscular pressure rather than shear wave velocity.
Reply: Direct measurement of intramuscular pressure of the leg muscles has potential risk to invade skin and muscles. Then, we adopted alternative measurement that can estimate intramuscular pressure with non-invasive method. In our previous study, we stated that shear wave elastography estimated intra-compartmental pressure of the lower leg muscles. This content is added in Research Limitations.
No direct intramuscular pressure measurement is performed.
In this measurement, the internal pressure of the leg muscle compartment was estimated as the change in the value of the shear wave velocity. For more direct measurement, intramuscular pressure measurement is required [46,47].
- "velocity" and "speed" are mixed throughout the paper. The title and table of the paper states, "shear wave speed" and the text states "shear wave velocity". It must be unified into one.
Reply: Word inconsistency is addressed correctly, at the Title, Key words, and Table2.
- Sample size part (the number of participants) should be in the “subjects” section in methodology.
Reply: Sample size statement is moved to subjects section in methodology.
- It would be better to present the reliability and validity of the "Ultrasound system (Aplio 500 Canon) and 6ì¸ wide linear probe (PLT100BT5)" equipment and measurement method.
Reply: Canon corporation has not provided the reliability and validity data of measuring SWV in previous literature. However, data published in PLosOne by Suh CH et al 2019 stated the accuracy of the velocity measurement by Aplio500 using a phantom (CIRS, Norfork, Virginia, USA; model 049 and 049A) of 8.0 ± 3.0 kPa is 0.49 kPa, and the precision (Coefficient of Variation and Confidence Interval) is 6.96%, 5.79–8.13%. In their study, validation of shear wave velocities measured by 5 major ultrasound systems (VTQ, VTIQ, EPIQ 5, Aixplorer, and Aplio 500) was performed using various phantoms with mutual comparison. The amount of measurement errors by Aplio 500 was minimum among them. We believe that the validity of the "Ultrasound system (Aplio 500 Canon) and 58 mm wide linear probe (PLT100BT5)" equipment is assured. At line 184-194.
Accuracy of SWV
We refer to a report in which measurements were made with an ultrasound system (Aplio 500, Canon, Tokyo, Japan) and a linear probe (PLT100BT5, Canon, Tokyo, Japan) with a field of view of 58 mm and a standard operating frequency of 10 MHz (maximum 14 MHz), and measurement accuracy was evaluated using phantoms [31]. According to the report, the accuracy of the velocity measurement by Aplio500 using a phantom (CIRS, Norfork, Virginia, USA; model 049 and 049A) of 8.0 ± 3.0 kPa is 0.49 kPa, and the precision (Coefficient of Variation and Confidence Interval) is 6.96%, 5.79–8.13%. It is known that the accuracy and precision of SWV measurements in the range of 2-6 cm depth of phantom material does not differ with depth [31]. In their study, validation of SWV measured by 5 major ultrasound systems (VTQ, VTIQ, EPIQ 5, Aixplorer, and Aplio 500) was performed using various phantoms with mutual comparison. The amount of measurement errors by Aplio 500 was minimum among them. The range of leg muscle SWV measured by the Aplio 500 in this experiment was 1.0-3.0 m/s (3.0-9.0 kPa), so the reliability and validity of the measurement of the SWV of the leg muscles by Aplio 500 was assured [21].
- Line 97. "Table 1" is marked as duplicate.
Reply: Duplicated part is removed, at Line 97.
- Why did you distinguish "Shear Wave Velocity" from "The Rate of Change in Shear Wave Velocity" in the results? The two do not seem to mean much differently. Also, even the table presented in the conclusion is the same.
Reply: “The Rate of Change in Shear Wave Velocity” is removed from Results and Discussion, and Table 3 is removed based on suggestions, at Line 256-272.
- Line 124: “a, 1b” à “1a, 1b” & Line 125, 126: a à 1a, b à 1b
Reply: Tipo errors are corrected in the text as suggested, at Line 124 and 125.
- Line 150: “a. 3b” à “3a, 3b” & Line 151, 152: a à 3a, b à 3b
Reply: Tipo errors are corrected in the text as suggested, at Line 150.
- In the conclusion section, raising of the lower leg decreases the intra-partial pressure by reducing the "connect issue water". However, this was not sufficiently presented in the discussion. It was presented only on line 366-368. This is too reasoning and leap forward to draw the conclusions presented by the author.
Reply: Overstatement in the conclusion is corrected as suggested, at Line 379 and 381.
Conclusions
Considering the changes in SWV and leg circumference associated with prolonged sitting, it is estimated that sitting legs causes an increase in intramuscular compartment pressure and increase in intravascular blood volume with retention of subcutaneous fluid in the lower leg.
Submission Date
05 May 2022
Date of this review
27 Jun 2022 04:59:02

Round 2
Reviewer 2 Report
All of those contents have been well revised.
Thank you for your hard work.
Good luck!